# MacGyvered Multiproperty Materials Using Nanocarbon and Jam: A Spectroscopic, Electromagnetic, and Rheological Investigation

**DOI:** 10.3390/jfb13010005

**Published:** 2022-01-10

**Authors:** Antonino Cataldo, Matteo La Pietra, Leonardo Zappelli, Davide Mencarelli, Luca Pierantoni, Stefano Bellucci

**Affiliations:** 1ENEA Centro Ricerche Casaccia, DISPREV Santa Maria di Galeria, 00123 Rome, Italy; 2Department of Information Engineering, Polytechnic University of Marche, 60131 Ancona, Italy; l.zappelli@staff.univpm.it (L.Z.); d.mencarelli@staff.univpm.it (D.M.); l.pierantoni@staff.univpm.it (L.P.); 3National Institute of Nuclear Physics (INFN), National Laboratories of Frascati, 00044 Frascati, Italy; matteo.lapietra.97@gmail.com (M.L.P.); Stefano.Bellucci@lnf.infn.it (S.B.)

**Keywords:** hydrogel nanocomposites, rheological characterization, electrical characterization, carbon nanostructures

## Abstract

As part of a biopolymer matrix, pectin was investigated to obtain an engineered jam, due to its biodegradability. Only a few examples of pectin-based nanocomposites are present in the literature, and even fewer such bionanocomposites utilize nanocarbon as a filler—mostly for use in food packaging. In the present paper, ecofriendly nanocomposites made from household reagents and displaying multiple properties are presented. In particular, the electrical behavior and viscoelastic properties of a commercial jam were modulated by loading the jam with carbon nanotubes and graphene nanoplates. A new nanocomposite class based on commercial jam was studied, estimating the percolation threshold for each filler. The electrical characterization and the rheological measurements suggest that the behavior above the percolation threshold is influenced by the different morphology—i.e., one-dimensional or two-dimensional—of the fillers. These outcomes encourage further studies on the use of household materials in producing advanced and innovative materials, in order to reduce the environmental impact of new technologies, without giving up advanced devices endowed with different physical properties.

## 1. Introduction

Could you MacGyver an advanced functional material at home using reagents from your fridge or your pantry? The well-known main character of the 1980s TV series was able to find outstanding solutions with everyday objects, such as paperclips, chewing gum, or rubber bands. Due to these extraordinary skills, MacGyver became synonymous with finding the solution to any problem with simple things, and was included in the *Oxford English Dictionary* as a verb, meaning “to make, form, or repair (something) with what is conveniently on hand.” [1]. Moreover, what could be better than MacGyvering a multiproperty material using jam as a starting point?

Jam is a “composite” in which the filler is the juice and flesh of a fruit, and the matrix is composed of water, sucrose, and pectin [2]. If one changes the type of filler—e.g., metal particles (copper, silver, aluminum, etc.), their oxides (titanium dioxide, zinc oxide, aluminum oxide, etc.), and carbon fillers (carbon black, carbon nanotubes, graphene, etc.)—one can obtain multifunctional properties for bio-based composites. In particular, hydrogels, prepared from natural polymers, have aroused interest due to their safe nature, biocompatibility, hydrophilic properties, and biodegradability [3]—and especially for their potential applications, such as bioconductors, biosensors, bioactuators, electro-stimulated drug delivery systems, and neuron-, muscle-, and skin-tissue engineering [4,5,6,7,8].

Among the natural polymers, chitosan and alginate have successfully been used in many fields [9,10,11,12,13]. They have been applied in environmental applications, as a matrix to remove pollutants, and extensively used in medical applications, such as for bioprinting and tissue engineering in regenerative medicine, as well as in drug delivery systems, as carriers for all kinds of drugs (e.g., proteins, antibiotics, vaccines) via different routes of administration, ranging from oral to nasal to cerebral [14,15,16]. Additionally, chitosan and alginate derivatives have been tested in order to verify their antioxidant and anti-inflammatory activities [17,18,19,20]; it has been demonstrated that these polysaccharides possess especially strong antioxidant activity, and their derivatives with polyphenols in particular induce an important antioxidant response.

As part of a biopolymer matrix, pectin was investigated to obtain an engineered jam, due to its biodegradability, related to its degree of esterification and polymerization [21].

Some examples of pectin-based nanocomposites are present in the literature. Bionanocomposites filled with ionic solids (such as iron oxide and thorium(IV) tungstomolybdate) have been studied in environmental applications: the selective removal of heavy metals [22] (Cu) or ionic dye [23] and the photodegradation of organic substances (methylene blue) [24] were both successfully tested. Furthermore, sustainable nanocomposites were obtained using nanoclay as a filler [25,26]. Exploring the bionanocomposites with nanocarbon as a filler, few examples are present, to the best of our knowledge, and those that exist studied the mechanical properties of nanocomposite films [27,28] for use in food packaging.

Therefore, the use of a pectin matrix starting from a commercial product, in our opinion, has the advantage of being a cheap and easily available solution, and represents a first attempt to transform a household material into an advanced one with tunable properties.

Beyond materials obtained from common jam, few papers have applied household solutions to pursue scientific results: only three papers in the Scopus database contain the words “household reagents” in the title [29,30,31]; among these, just one, by Peper et al. [31], used reagents that can be purchased in a supermarket. To the best of our knowledge, findings regarding inkjet printing using a commercial inkjet printer [32,33] and exfoliation of graphite using Coca-Cola^®^ or black tea [34,35] are other field applications of household solutions.

In order to exploit the safe nature of the matrix and the low cost of the reagents, we wanted to explore the possibility of producing advanced materials using a carbon nanostructure as a filler, and to compare the effects of different dimensionality—1D or 2D. To this end, carbon nanotubes (CNTs) and graphene nanoplates (GNPs) were used. Carbon structures as filler materials are currently proposed in several fields of application—for instance, to reinforce metals [36], carbon fiber polymer composites [37], and polylactide (PLA) [38]. Moreover, their use in the medical field has been investigated—CNTs and graphene have been extensively studied for use in drug delivery systems [39,40,41] and tissue engineering [42,43,44]; moreover the electrical properties of CNTs have been applied to connect neuronal cells [45].

Additionally, to reduce the environmental impact, the GNPs were produced using an eco-friendly method developed in previous works [46,47,48]. Finally, to demonstrate the ability to produce advanced materials using household reagents, the matrix was obtained starting from a jam formulation purchased in a supermarket.

Thus, in this paper, we propose the preparation of an advanced functional material using jam as a matrix. In addition to the MacGyvered process, the use of jam as a matrix is an opportunity to use a green and cheap material to fabricate advanced devices. A robust electrical characterization, supported by spectroscopic and viscoelastic investigations, was carried out.

## 2. Experimental

### 2.1. Materials

In this work, both pectin and sugar (sucrose, purity 100%) were purchased from a supermarket; specifically, S. Martino^®^ “Frutti in Casa 2:1 “pectin formulation (https://www.ilovesanmartino.it/catalogo/cioccolata-e-marmellata/marmellate-e-confetture/fruttincasa-2-1/, accessed on 18 November 2021) was used. Multiwalled CNTs (diameter 8–15 nm, average length 50 µm, purity > 99 wt%, cod M2702, Heji, Shenzhen, China) produced via chemical vapor deposition were purchased from HeJi. Graphene nanoplates (GNPs) were produced in the laboratory by microwave irradiation, starting from Asbury^®^ expandable graphite (purity 99.1%, cod EG 1721, Asbury Graphite Mills Inc, Asbury, IA, USA). The synthesis method used in this work, developed by the NEXT Nanotechnology Group (INFN-LNF) [46,47,48], involves the use of EG with SO_4_^2−^ and NO_3_^−^ as intercalating agents. The exfoliation process took place in a household microwave oven (thus replacing the furnace, in which it would reach a temperature of ~1000 °C) as follows: 1–2 g of EG was placed in a ceramic pot, which was placed in the microwave oven and irradiated at 850 W for few seconds. After irradiation, expanded graphite with a worm-like structure was obtained. These structures were sonicated in isopropyl alcohol (RS PRO, purity 99.7%) for ~1 h, crumbling the worm-like particles and causing GNPs to appear. The GNP powder was subsequently filtered to recover any alcohol, and then placed in an oven at 45 °C to evaporate the solvent residues. The characteristic structure of pectin is a linear chain of alpha(1-4)-linked d-galacturonic acid that forms the pectin backbone—a homogalacturonan, as shown in Figure 1.

### 2.2. Hydrogel Preparation

The jam hydrogel was obtained by mixing pectin and sugar in distilled water.

Fifty milliliters of 20% (*w*/*v*) of sugar solution was stirred with a magnetic stirrer until complete solubilization. Then, 4 g of pectin was added to the solution and completely solubilized, and the solution was placed on a heated plate to bring it to the boil. Sugar decreased the hydration of the pectin by competing for water, as a result of which the stability of pectin decreased. During the cooling, the hydrogen bonding and hydrophobic interaction caused the aggregation of pectin chains [50]. Finally, the solution was boiled for ~5 s and then left to rest for a day.

The nanocomposites were prepared by adding the nanocarbon fillers (GNPs and CNTs) to the sugar and pectin solution, which was brought to the boil and left to rest for a day. The resulting nanocomposites were characterized by a poor dispersion of filler inside the matrix. This problem was solved by adding a sonication step to the procedure; thus, we prepared a pectin–filler solution, which was sonicated using an ultrasonic tip (Sonics Vibra-Cell, 40 MHz, Sonics & Materials Inc, Newtown, CT, USA) with an amplitude of 40% for 60 s. The pectin played the role of a dispersant, stabilizing the nanostructures. After complete dispersion, the sugar was added to the solution, which was brought to the boil and left to rest for a day.

As an example, a picture of the 0.05% GNP/pectin hydrogel is shown in Figure 2.

## 3. Characterization

The pure matrix and nanocomposites were characterized using Raman spectroscopy (Renishaw, Invia, Wotton-under-Edge, UK), as well as impedance and rheological measurements.

Raman spectra were recorded using an inVia Raman microscope (Renishaw), equipped with a 532 nm laser and a 600 l/mm grating.

Raman characterization was performed on freeze-dried samples, because water would have disturbed measurement. The other settings used were as follows: laser power of 50%, exposure time of 2.5 s, and 20 accumulations [15].

Impedance measurements were performed using an MFIA 5 MHz Impedance Analyzer (Zurich Instruments, Zurich, Switzerland). The measurements were carried out by setting a frequency range from 10 Hz to 1 MHz.

The specimens were placed in a sample holder (Figure 3), which was placed between two gold electrodes. The setup was calibrated using the short/open method—the fixed distance between the electrodes guaranteed a good reproducibility of the measurements. The gold electrodes were connected to the instrument with Kelvin clamps (fixing the geometry of the cables), and then the short/open measurements were collected. Finally, the electrical measurements of the samples were carried out. The electrical measurements were used to obtain the lumped circuit, as reported in the literature [51,52].

Rheological measurements were obtained using an Anton Paar Modular Compact Rheometer 302 (MCR) (Anton Paar, Gratz, Austria) with a 25 mm plane plate configuration. An amplitude sweep test was performed using an angular frequency of 10 rad/s [15].

## 4. Results and Discussion

### 4.1. Raman Spectroscopy

The Raman spectra of the jam, 0.05% CNT (blue), and 1.5% CNT (red) composites are shown in Figure 4.

The pure jam and 0.05% CNT specimens show the same Raman pattern, even if the low-intensity peaks characteristic of CNTs are visible. The 1.5% CNT specimen has a spectral pattern completely different from the others, showing mainly the characteristic bands of nanocarbon compounds: a D band around 1350 cm^−1^, a G band around 1580 cm^−1^, a 2D band around 2700 cm^−1^, and a peak around 4300 cm^−1^ related to the presence of nanocarbon. In all of the spectra, the peak related to water, at around 3400 cm^−1^, exhibits a slight red shift, increasing the CNT content.

The Raman spectra of the jam, 0.05% GNP (blue) and 1.5% GNP (red) composites are shown in Figure 5.

In the case of jam loaded with GNPs, the Raman spectrum results of pure jam were not modified by the loading of nanocarbon. The 1.5% GNP specimen’s spectrum presents marked bands of nanocarbon compounds: a G band around 1580 cm^−1^, and a 2D band around 2700 cm^−1^. These data also show a red shift of the water peak with increasing GNP content. The common behavior of the CNT and GNP specimens suggests the same effect of nanocarbon inside the jam matrix—during the gelling process, the presence of sugar is fundamental to reduce the activity of water and to stabilize junction zones by promoting hydrophobic interactions [21]. The nanocarbon, having a hydrophobic nature, could help the stabilization via hydrophobic interaction; consequently, the structure becomes more compact and the hydrogen bonds become stronger, as suggested by the red shift [53].

### 4.2. Impedance Measurement

The graphs in Figure 6a,b, show the amplitude and phase of the measured impedance (Z) of the CNT/jam nanocomposites.

It is possible to observe the decrease in the impedance values (Figure 6a) with the increasing concentration, which is more evident at low frequencies. Moreover, the pattern of phases (Figure 6b) also shows an evolution—the phase pattern of pure jam shows a capacitive resonance peak around 100 Hz that moves towards lower frequencies with increasing filler concentration, falling below 10 Hz for 1% and 1.5% filler concentrations. The marked modification in the phase behavior between the 0.75 and 1% CNT samples suggests the exceeding of the percolation threshold.

The graphs in Figure 6c,d, show the amplitude and phase of the measured impedance (Z) of the GNP/jam nanocomposites.

As in the CNT nanocomposite specimens, it is possible to observe the decrease in impedance values with the increasing concentration, which is more evident at low frequencies, corresponding to an evolution in the phase pattern; a shift in the capacitive resonance peak (at 100 Hz) is also visible in the GNP nanocomposite series, but only for the 1.5% sample is it lower than 10 Hz. Moreover, it is possible to observe the emergence of a new resonance peak in the region of high frequencies, ~100 kHz, which was not observed in the CNT nanocomposites.

Because of the marked modification in the phase behavior at the highest concentration, we suggest that the exceeding of percolation threshold takes place between 1 and 1.5% GNP concentration.

To complete the electrical characterization, the fitting with an electrical lumped circuit was performed. In order to find the best fitting, circuits with a resistance in series to RC parallel groups were tested (up to four RC groups; Figure 7). The R∞_,_ in series to the RC parallel elements represents the resistance contact with the interface electrode/composite.

The best fitting occurred with the lumped circuit with the R∞ in series to four RC parallel groups.

The r^2^ increased progressively with the increase in the number of RC groups, reaching the highest values with four RC groups for both the module and phase of impedance (all of the values are present in the Appendix A). The results of the fitting are reported below, comparing resistances R_∞_, R_1_, and capacitance C_1_ in terms of the concentration of each filler (Figure 8).

The addition of filler to the matrix produced a marked decrease in resistances even with a tiny amount of filler—0.05% for CNT and 0.25% for GNP fillers. Meanwhile, the capacitance showed a good improvement for the same amount of filler. Similar behaviors were also observed for the other resistances (R_2_, R_3_, R_4_) and capacitances (C_2_, C_3_, C_4_).

Next, we used the parameters obtained by the fitting to simulate and reconstruct the phase pattern—especially for specimens in which the resonant peaks were below the lower frequency limit (10 Hz). In order to check the accuracy of the simulation, we plotted the pure matrix—empty jam (black), which presents the peak—as well as CNTs and GNPs at concentrations of 1 and 1.5% for each filler. Astonishingly, the simulated curves reproduced the behavior of the phase pattern, especially at low frequencies.

In the simulated curves for the CNT composites (Figure 9a), the dashed lines reproduce the phase curve perfectly, enabling determination of the resonant frequency for the 1 and 1.5% CNT samples, and confirming the resonant peak for the jam specimen.

In the simulated curves for the GNP composites (Figure 9b), the dashed lines reproduce the phase curve perfectly, enabling determination of the resonant frequency for the 1.5% GNP sample, and confirming the resonant peaks for the jam and 1% GNP specimens, as in the case of CNT composites.

Moreover, the simulation predicted the emergence of a second resonant peak for GNP specimens, even if the simulated peak was placed at the lowest frequency with respect to the experimental peak. It should be noted that the prediction of the second peak could be improved by adding other RC groups to the equivalent circuit. The details of the resonant frequency are reported in Table 1 and Table 2.

These results reinforce the estimation of the exceeding of the percolation threshold between 0.75 and 1% and between 1 and 1.5% for the CNT and GNP fillers, respectively.

### 4.3. Rheological Measurements

The graph reported in Figure 10 shows the trend of the storage modulus (G’) and the loss modulus (G”) as a function of shear strain of the jam–CNT nanocomposites. According to ISO 6721-10, we calculated the limit of the linear viscoelastic (LVE) region when a difference of 5% in G’ values occurred. The LVE region decreased gradually with increasing filler content. Since the LVE region is the limit at which elastic deformation occurs, we would argue that the presence of CNTs reduces the LVE region, because the movement of the matrix around the fillers causes irreversible deformations, which are much larger the higher the filler concentration is [54].

The nanocomposites, which show a solid-like behavior (G’ > G”), exhibit changes in the storage and loss modulus values. The G’ values increase with increasing filler content, highlighting the contribution of the rigidity of CNTs to the matrix. Moreover, the G” curves show an evolution in the pattern, in that the G” pattern of the jam specimen shows a maximum before the flow point; in the yield zone, between the LVE limit and the flow point, the sample starts to suffer inelastic deformation due to microcracks and internal friction. The growth of microcracks takes place up to the maximum G” to form a macrocrack, and the sample starts to flow at the flow point, very close to the maximum G”. From Figure 10, we can see the reduction and disappearance of the maxima of the G” curves due to the presence of CNTs, highlighting the formation of the CNT network within the matrix; in fact, the G” values, which describe the portion of the deformation energy that is lost to internal friction during shearing, increase with increasing filler content, suggesting an increase in rigidity due to the presence of CNTs and the formation of the CNT network [54].

This suggestion is confirmed by the Cole–Cole diagrams (Figure 11) in which the matrix relaxation mode of the jam (black curve) can be observed by the modification of the curve, along with the presence of a second relaxation mode due to the network above the percolation threshold.

Finally, we would remark that the presence of a large gap between the 0.75% CNT and 1% CNT nanocomposites, along with the change in the pattern of the Cole–Cole plot, is due to the exceeding of the rheological percolation threshold, which coincides with the electric percolation threshold.

The graph reported in Figure 12 shows the trend of G’ and G” as a function of the shear strain of the jam–GNP nanocomposites.

As in the case of the CNT/jam nanocomposites, the LVE region decreases gradually with increasing filler content, and the nanocomposites show a solid-like behavior (G’ > G”). Moreover, the G’ and G” patterns exhibit the same behavior: the G’ values increase with increasing filler content, showing a greater resistance to deformation of the jam/GNP nanocomposites. In the G” curves, it is possible to see the presence of the maximum before the flow point, highlighting the growth of microcracks. In Figure 12, we can see the disappearance of the peak in the G” pattern above 0.8% jam/GNP nanocomposite. This can be attributed to the formation of the GNP network inside the jam matrix, thus exceeding the percolation threshold. This suggestion is confirmed by the Cole–Cole diagrams (Figure 13), in which we can see the presence of a second relaxation mode due to the formation of the GNP network.

Thus, the rheological percolation threshold for GNP/jam specimens was estimated between 0.5 and 0.8%—lower than the electrical percolation threshold.

The difference between the rheological and electrical percolation thresholds can be explained using the reptation theory; in fact, the rheological percolation threshold is usually exceeded when the filler–filler distance is between the polymer entanglement distance and twice the radius of the gyration of the polymer. In the case of GNPs, their large surface area increases the number of entanglements at a larger distance, while the tubular surface of CNTs does so at a shorter distance. This distance is enough for GNP/jam specimens to exceed the rheological percolation threshold, but not enough to reach the electrical threshold [55].

This difference is highlighted in the trend of the flow point and in the yield zone (Figure 14a,b); the flow point tendency for CNT nanocomposites shows a break point at the minimum distance between fillers (according to the percolation threshold). The increase in concentration leads to an increase in the rigidity, corresponding to high flow point values. In the case of GNPs, the trend is monotonically descending, reflecting the fact that at the percolation threshold the distance is quite large, and an increase in filler concentration does not produce an increase in flow point values.

Moreover, for both GNP and CNT composites, the yield zone is wider than the yield zone of pure jam, but with different effects: the GNPs produce a lubricant effect, while CNTs do the opposite.

## 5. Conclusions

In the present work, we demonstrated the capability to obtain multiproperty materials via a MacGyvered approach, using household materials, in order to reduce the environmental impact and obtain a valuable low-cost material.

The two series of nanocomposites were made using jam (pectin/sugar hydrogel) as the matrix and nanocarbon (CNTs and GNPs) as a filler.

The Raman spectroscopy showed a red shift in the peak related to water, suggesting participation of nanostructures in the gelling mechanism, increasing the strength of hydrogen bonds due to their hydrophobic nature. Moreover, the electrical characterization highlighted a decrease in impedance values and a shift in the resonant peak—exhibited by pure jam at ~70 Hz—at lower frequencies. The electrical lumped circuit allowed us to predict the resonance peaks with a good approximation to the experimental ones. Moreover, the two fillers showed different behavior: the CNT/jam nanocomposites showed only the shift in the resonant peak, while the GNP/jam nanocomposites showed both the shift in the resonant peak and the appearance of a new resonant peak at the highest frequencies (~50 kHz). Finally, the rheological investigation showed a change in the viscoelastic properties and the different effects that the two nanostructures produced; if the networks of both nanostructures increased the elastic properties (G’ values), as suggested by Raman spectroscopy, the CNT/jam nanocomposites showed more rigidity to the deformation, as the high flow point values reveal, while the GNP/jam nanocomposites showed a lubricant effect, with low flow point values.

This work represents a good starting point to explore multiproperty materials made with household reagents. Further investigation should address the optimization of parameters—such as concentration, type of filler, and type of matrix—in order to modulate the electric and viscoelastic properties to produce appropriate devices. Taking advantage of electrical properties, these nanocomposites could operate as sensors for environmental pollutants. In our previous works, we produced a graphene-based device able to retain and detect organic pollutants in water [56,57,58]. Furthermore, by modulating the rheological properties, the nanocomposites could play a role as an epidermal drug release system. It is possible to release an epidermal therapy for long time with a single application, loading the hydrogel nanocomposite with specific drugs [15]. Furthermore, a proper investigation of biocompatibility must be carried out, although nanocarbon drug delivery systems have been extensively tested, and the same nanocarbon materials used in this paper were tested as drug delivery systems by our group, and we observed good biocompatibility, which was increased by a proper stabilizer [14,15,59].

## Figures and Tables

**Figure 1 jfb-13-00005-f001:**
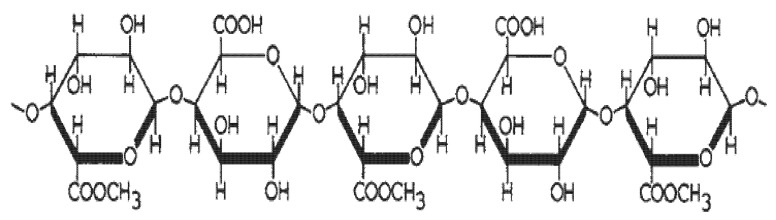
Pectin backbone structure, adapted from [49].

**Figure 2 jfb-13-00005-f002:**
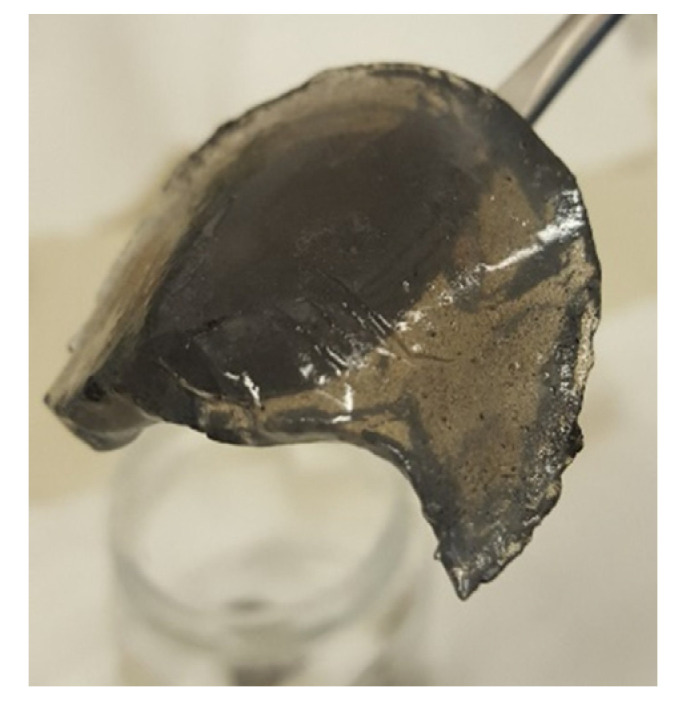
Picture of the 0.05% GNP/pectin hydrogel.

**Figure 3 jfb-13-00005-f003:**
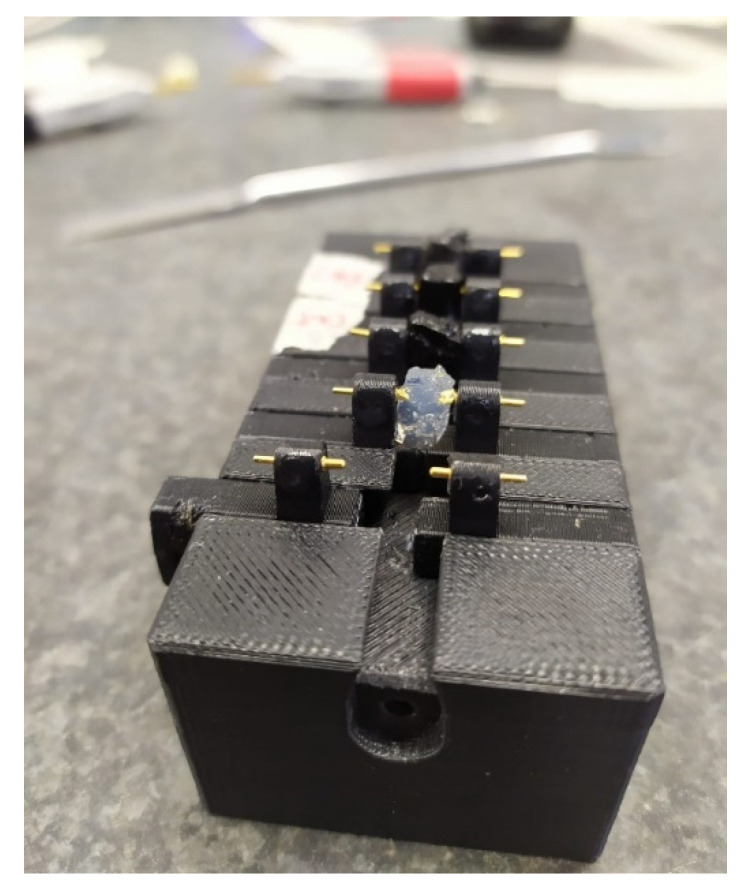
The sample holder used for the electrical characterization. The gold electrodes, placed at a distance of 2 mm, were useful to hold the specimen and simultaneously obtain the electrical characterization. The gold electrodes were connected to the impedance analyzer by means of Kelvin clamps.

**Figure 4 jfb-13-00005-f004:**
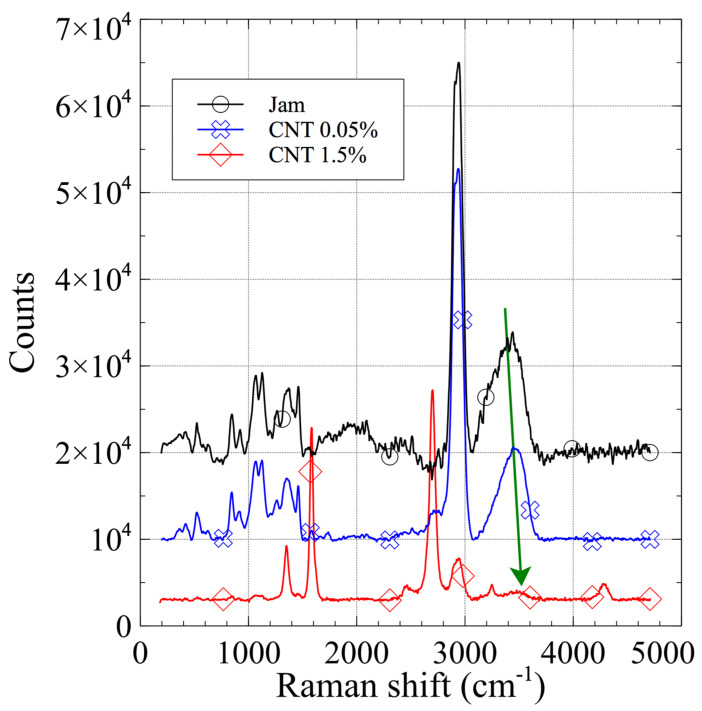
Raman spectra of pure matrix jam (black), CNT/jam 0.05% (blue), and CNT/jam 1.5% (red). The red shift of the peak related to water (4300 cm^−1^) suggests a slight change in the hydrogel structure due to the presence of CNTs.

**Figure 5 jfb-13-00005-f005:**
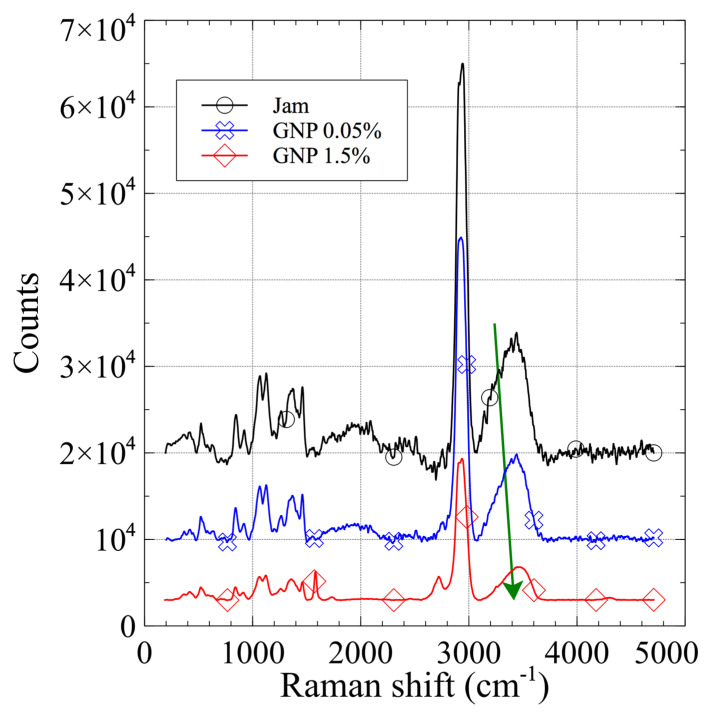
Raman spectra of pure matrix jam (black), GNP/jam 0.05% (blue), and GNP/jam 1.5% (red). The red shift of the peak related to water (4300 cm^−1^) suggests a slight change in the hydrogel structure due to the presence of GNPs.

**Figure 6 jfb-13-00005-f006:**
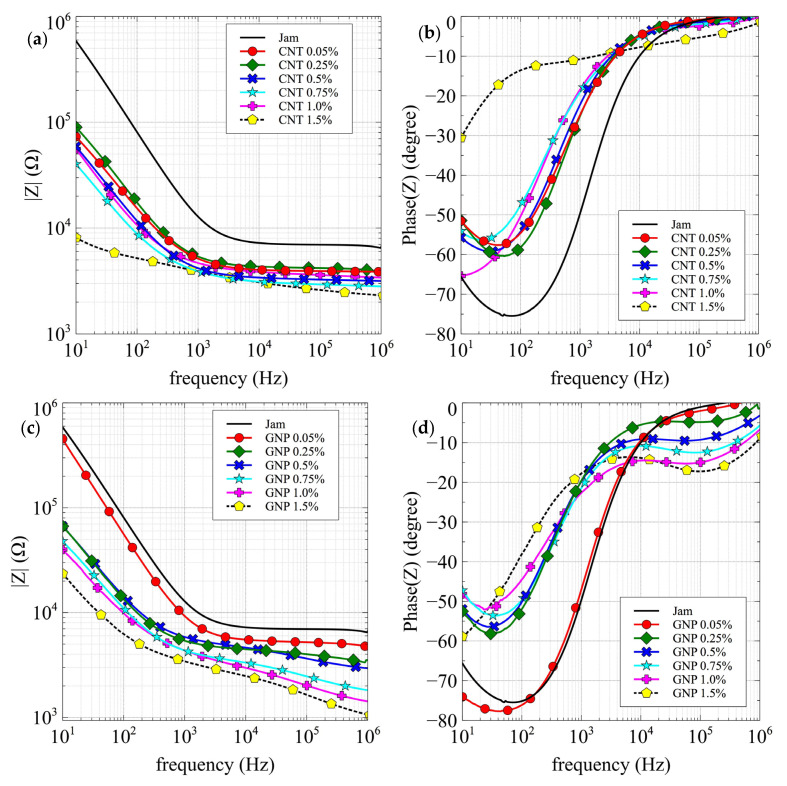
Impedance amplitude and phase vs. frequency: (**a**,**b**) jam and CNT nanocomposites; (**c**,**d**) jam and GNP nanocomposites.

**Figure 7 jfb-13-00005-f007:**
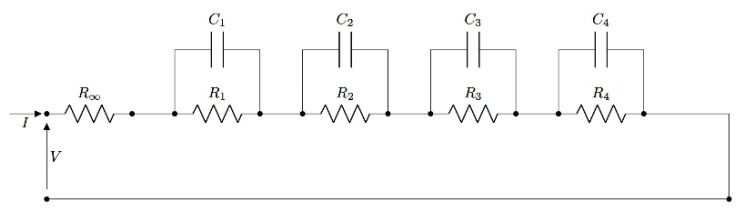
Lumped circuit representation with resistance in series to one and four RC parallel groups.

**Figure 8 jfb-13-00005-f008:**
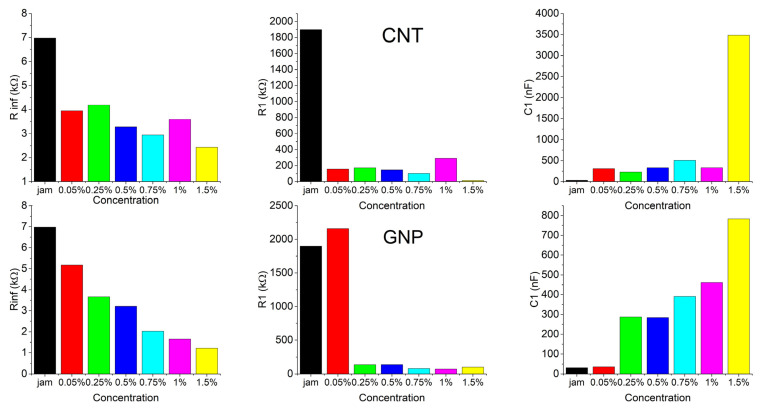
Best fitting results of R_∞_, R_1_, and C_1_ for CNT (**top**) and GNP (**bottom**) jam nanocomposites.

**Figure 9 jfb-13-00005-f009:**
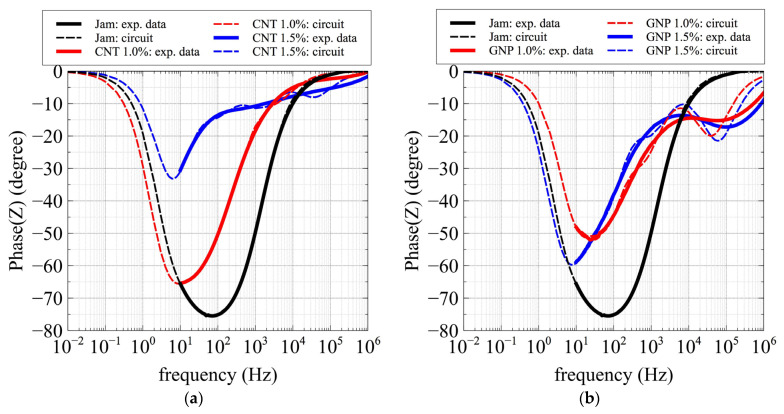
Comparison of the impedance phase patterns between the experimental data (solid lines) and the values obtained by the equivalent circuits (dotted lines) for pure jam (black) and (**a**) CNT/jam 1% (red) and CNT/jam 1.5% (blue); and (**b**) GNP/jam 1% (red) and GNP/jam 1.5% (blue).

**Figure 10 jfb-13-00005-f010:**
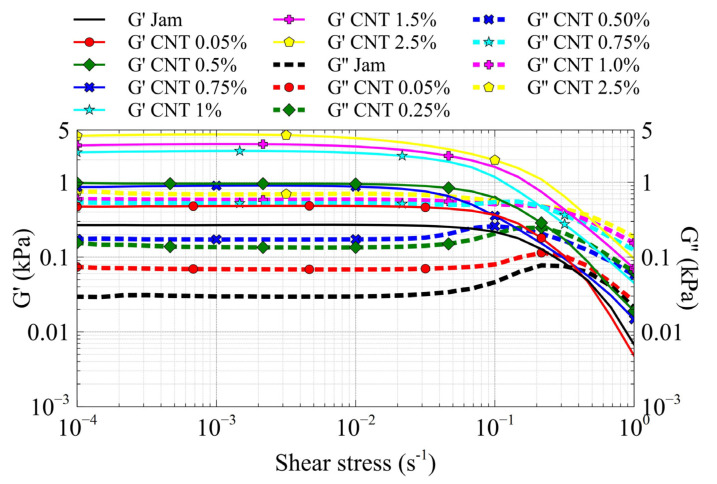
G’ (black) and G” (red) of the CNT/jam nanocomposites.

**Figure 11 jfb-13-00005-f011:**
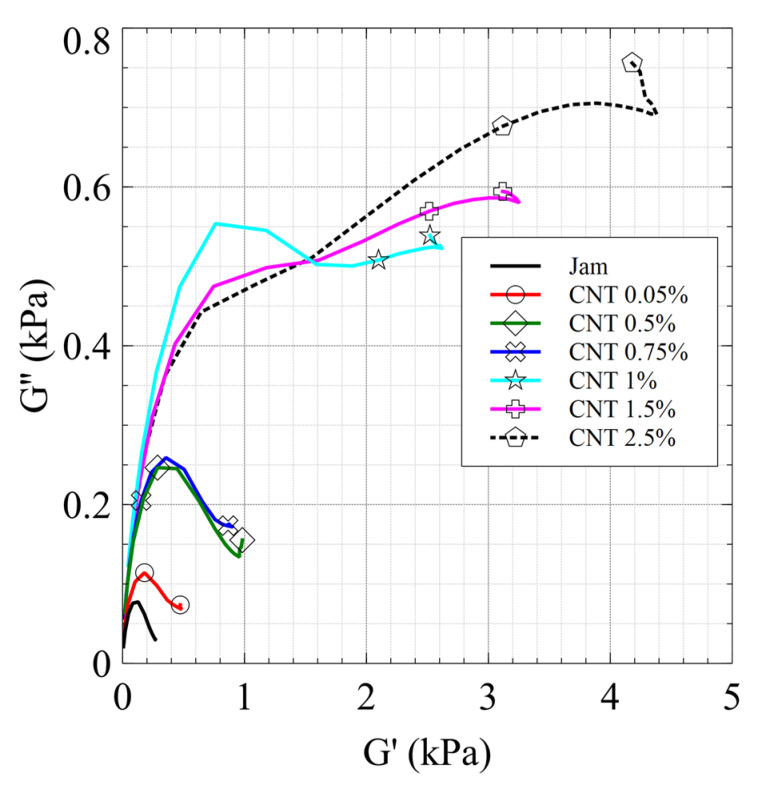
Cole–Cole diagram of the CNT/jam nanocomposites.

**Figure 12 jfb-13-00005-f012:**
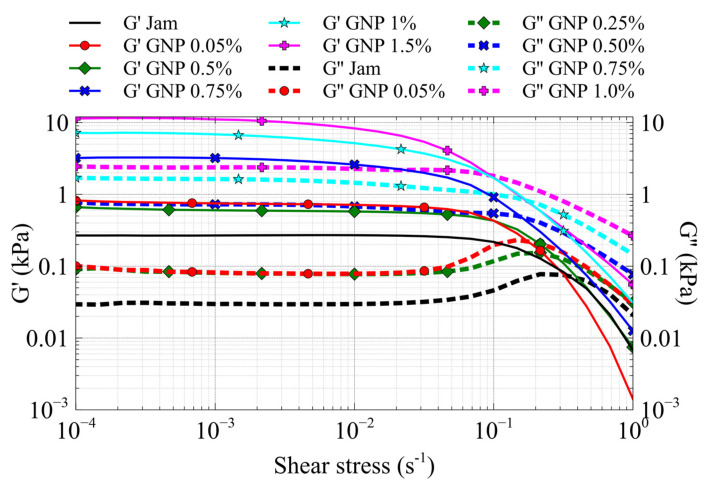
G’ (black) and G” (red) of the GNP/jam nanocomposites.

**Figure 13 jfb-13-00005-f013:**
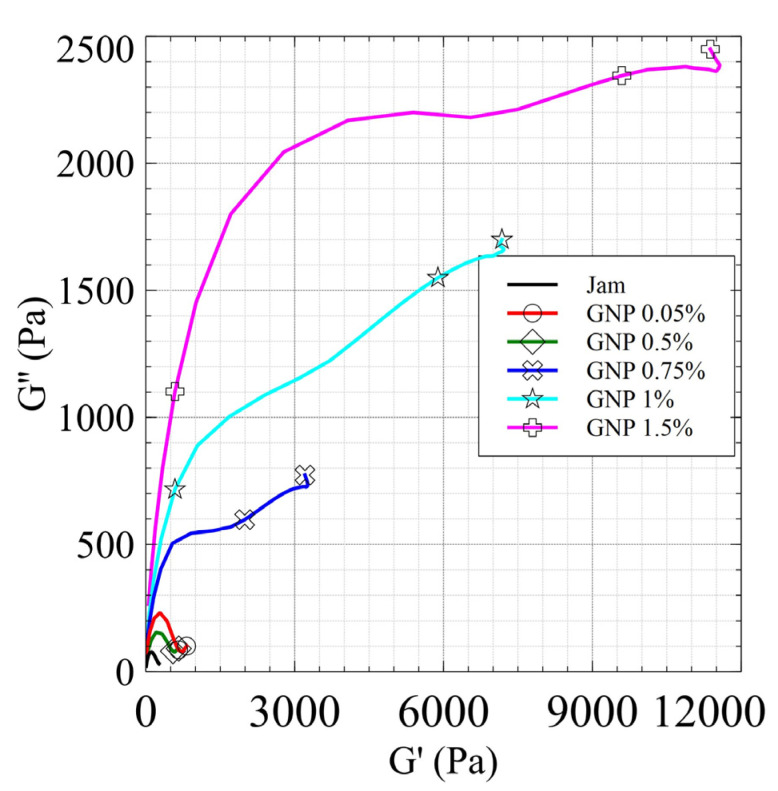
Cole–Cole diagram of the GNP/jam nanocomposites.

**Figure 14 jfb-13-00005-f014:**
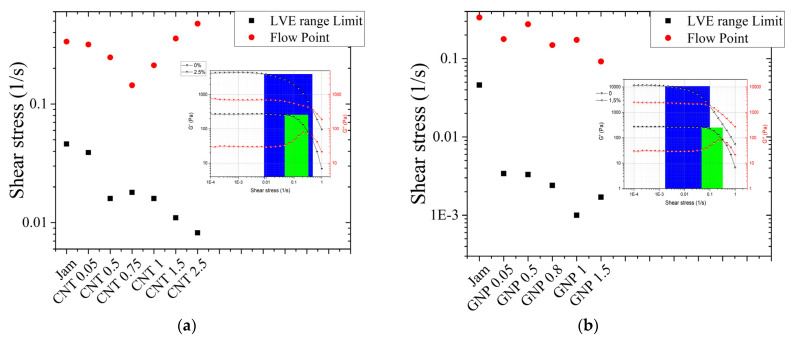
LVE range limits (black squares) and flow points (red dots) vs. concentration for (**a**) CNT/jam nanocomposites and (**b**) GNP/jam nanocomposites. In the insets, the representation of the yield zones (inset) for jam (green area) and the 2.5% nanocomposite (blue area) is reported. The black dotted and red dotted curves are the G’ and G” of pure jam and (**a**) CNT 2.5% and (**b**) GNP 1.5%, as reported in Figure 10 and Figure 12, respectively.

**Table 1 jfb-13-00005-t001:** Peak frequencies for CNT/jam nanocomposites in terms of concentration, along with simulated values. It is clear that the lumped circuit effectively reproduces the experimental resonance peaks.

Sample	Peak Frequency (Hz)	Simulated Peak Frequency (Hz)
Jam	69.0	71.5
CNT 0.05%	40.7	38.5
CNT 0.25%	53.0	46.9
CNT 0.5%	31.3	32.4
CNT 0.75%	22.0	24.1
CNT 1%	-	9.1
CNT 1.5%	-	6.3

**Table 2 jfb-13-00005-t002:** Peak frequencies for GNP/jam nanocomposites in terms of concentration, along with simulated values. It is clear that the lumped circuit effectively reproduces the experimental resonance peaks, and also predicts the second resonant peaks.

Sample	1st Peak Frequency (Hz)	2nd Peak Frequency (Hz)	1st Simulated Peak Frequency (Hz)	2nd Simulated Peak Frequency (Hz)
Jam	69.0	-	75.6	-
GNP 0.05%	53.0	-	47.2	-
GNP 0.25%	31.3	88,194.4	36.4	52,266.1
GNP 0.5%	27.4	56,864.3	36.3	40,128.8
GNP 0.75%	37.3	80,783.1	45.4	49,621.7
GNP 1%	24.1	56,864.3	21.5	38,453.2
GNP 1.5%	-	96,285.5	7.8	59,672.1

## Data Availability

The data presented in this study are available on request from the corresponding author. The data are not present in public databases.

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
