# Peer review of "MacGyvered Multiproperty Materials Using Nanocarbon and Jam: A Spectroscopic, Electromagnetic, and Rheological Investigation"

_jfb, 2022, doi:10.3390/jfb13010005_

Round 1
Reviewer 1 Report
The manuscript entitles “MacGyvered multifunctional materials using nanocarbon and 2
jam: a spectroscopic, electromagnetic and rheological investigation” investigated that the potential of multifunctional materials and their applications. The subject frame of the work is well constructed. So, in this respect and this article should be contributed to present research. I recommended this work for re-submission after the following revisions.
- There are several typographical mistakes as well in whole manuscript. Therefore, the author’s thoroughly careful check the language and typo mistake to minimize the error.
- The abstract should be beginning with a sentence about the background of concept and the aims as well as novelty of study should be mentions. What exactly is the novelty of this study? The abstract is poorly written and should be improved. Abbreviations must be avoided in abstract. Parenthesis should be avoided in abstract - this is poor writing. Please improve.
- Introduction; Check and format the citations in the whole manuscript. Also, Appropriate references must be provided to explained the background, what is already done and why this study carried out. Other vise the novelty of this research is still poorly presented. This is important especially for the high IF journals. The scientific style should be used. What exactly is the aim of this work? Hypothesis statement is missing in the introduction section.
- Introduction; Introduction section should be elaborated with recent articles on antioxidant and anti-inflammatory activities. Also about nano-carbon materials as well as the importance in various fields is missing.
- Material and methods; The whole M&M section is poorly written and must be substantially rewritten and improved. The methods are not properly referenced and are not possible to follow, reproduce and verify. This is serious drawback of this research work. You must be precise when writing protocols (M&M section); everyone should be able to repeat your experiments, after this paper is published, and gain the same results. Serious lack of relevant information or poorly written Materials and methods section are always an alarm form me, as a reviewer. For example, Materials are poorly described. Furity and product codes of all these materials should be provided. Materials are poorly described. Product codes should be provided. Appropriate references are missing. Who exactly is the Author of the methods applied? Are these methods valid/correct? Without sufficient information it is hard to reproduce the results, follow, and compare the idea and the obtained results with the literature. Appropriate references to the methods should be provided.
- The statistical findings have to be given in the text such as (p<0.05) or (p>0.05).
- Results and discussion; General remark to the discussion - In my opinion, the discussion provided by Authors is difficult to follow and verify due missing critical details in the methodology section. Due to poorly described material and poorly presented methods, I am not able to follow and properly review the discussion.
- All figures are of poor technical quality and not suitable for publication, especially in a high reputed journal. Font size and kind is too small and must be unified in all figures. Small writings are unreadable. All figures must be self-explanatory. Axis titles are poorly presented or absent. Units are missing. Are the data presented in figures significantly different? At least error bars should be shown.
- Details about the statistical testing are needed.
- I suggest first time write full name rather than abbreviation; revise throughout in manuscript
- Figure 1, is took it from literature need citation. if you draw by yourself need more explanation about bounding types etc.
- There were figures captions confusion. Please look at figure they were many time prepared. Please mark it figure 2(A), B, and C. same is for rest of figure. Its presented roughly.
- Figure 4 and 5 should be combine in to A and B; Same is for figure 8 and 9
- I did not understand about tables. you need to draw in it journal representation format. Whole data is roughly represented.
- All results especially figures and tables were not represented in proper way. I suggest carefully revised and re-submitte
Author Response
First of all, we apologize to the reviewer because the proof that they review was full of typos due to some problems during formatting operated by journal. Nevertheless, we appreciate the reviewer’s comments/suggestions and we did the best to improve the paper according to comments/suggestions.

Reviewer 2 Report
Author presented the use sugar and pectin to synthesized multifunctional hydrogel. This is a unique way to make such material and reduce environmental impacts of new technologies. However, I found the title and information in the text is somewhat missing therefore it cannot be published in its current form. Following comments may be helpful in rewriting the manuscript:
- The title says multifunctional materials however, its application was not presented/discussed in the text. Authors are suggested to use suitable title for the manuscript.
- The figures quality must be improved and suitably labelled. Also, some figure captions were missing, and some figures were not cited correctly or not even cited in the text. Please check line 84, 96, 108, 190, 221, 227.
- Authors are suggested to use figure numbers in chronological order and cite them in the text accordingly.
- The introduction can be improved by including more references of the current method (using jam or any other household reagent).
- In experimental section, please provide the detailed experimental steps, parameters, conditions and quantity used to obtained graphene nanoplatelets.
- Line 79, what does author meant by work-like structure? Is it a typo to ‘worm-like’ or something else?
- Author synthesized jam nanocomposite using CNT and GNP as fillers, what is the biocompatibility of final product after microwave exposure to graphite? Also, please discuss possible applications of the synthesized product.
- What does R∞ signifies on page 6?
- There are two table 1, please check.
- Please proofread to avoid grammatical and typo errors.
Author Response

(The authors gave the same response as above.)

Reviewer 3 Report
This work by Cataldo et al. is essentially interesting. They turned jam into conductive and elastic gel by compositing with carbon nanotubes and graphene nanoplatelets. There are still a few issues to be addressed in this current manuscript. Therefore, I recommend it to be published in Journal of Functional Biomaterials after a minor revision.
- The fonts of figure legends in several figures need to be adjusted: Figure 2,3, S6, S7.
- In the Raman spectra of jam with GNP and CNT, the relative intensity of 2D band from GNP is significantly lower than that from CNT. What is the possible reason?
- The SEM is not a suitable method to characterize such materials, the dimensions of either CNT of GNP are much smaller than the scale presented (10 um).
Author Response

(The authors gave the same response as above.)

Round 2
Reviewer 1 Report
No further comment
Author Response
Thanks to the reviewer. We are pleased that our review has allayed its initial perplexities
Reviewer 2 Report
Author made significant improvement based on the comments. I would suggest following comments to be addressed before its publication:
- Please delete ‘electrical’ from line 15, as commercial jam does not have electrical properties.
- Line 106, correct work-like or define what is work-like? Page 10, table 1, please correct the formatting error.
- Author mentioned possible application as epidermal drug release using synthesized hydrogel. It would have been better if author discusses a little bit about biocompatibility of the hydrogel for drug delivery systems. How to improve its shelf life to avoid mold growth?
- In supporting information, there are still two table 3. Please correct the table numbers.
Author Response
Thanks to reviewer for the comments.
1- Please delete ‘electrical’ from line 15, as commercial jam does not have electrical properties.
Thanks for the comment, but we are not agree with the suggestion to delete electrical. In fact, we demonstrate experimentally that the commercial jam has a behavior as resistor with a tiny contribute of capacitor. So we change electrical properties in electrical behavior.
2- Line 106, correct work-like or define what is work-like? Page 10, table 1, please correct the formatting error.
We correct work-like in worm-like. Furthermore, we change the style of table with the same of the text.
3- Author mentioned possible application as epidermal drug release using synthesized hydrogel. It would have been better if author discusses a little bit about biocompatibility of the hydrogel for drug delivery systems. How to improve its shelf life to avoid mold growth?
Thanks for the comments. We add a paragraph in which we discuss briefly about the biocompatibility of the specimen. In order to improve the shelf life, we can use the strategy to freeze-dry the specimens.
4- In supporting information, there are still two table 3. Please correct the table numbers.
We correct the order of the table in SI.